# Comprehensive Evaluation of Tomato Growth Status under Aerated Drip Irrigation Based on Critical Nitrogen Concentration and Nitrogen Nutrient Diagnosis

**DOI:** 10.3390/plants13020270

**Published:** 2024-01-17

**Authors:** Hongjun Lei, Yiming Fan, Zheyuan Xiao, Cuicui Jin, Yingying Chen, Hongwei Pan

**Affiliations:** School of Water Conservancy, North China University of Water Resources and Electric Power, Zhengzhou 450046, China; leihongjun@ncwu.edu.cn (H.L.); z20211010120@stu.ncwu.edu.cn (Y.F.); x201810102054@stu.ncwu.edu.cn (Z.X.); b2018081504@stu.ncwu.edu.cn (C.J.); chenyingying@ncwu.edu.cn (Y.C.)

**Keywords:** aerated drip irrigation, tomato, nitrogen uptake, critical nitrogen concentration, nitrogen nutrition index

## Abstract

In order to provide a theoretical basis for the rational application of nitrogen fertilizer for tomatoes under aerated drip irrigation, a model of the critical nitrogen dilution curve was established in this study, and the feasibility of the nitrogen nutrition index (*NNI*) for the real-time diagnosis and evaluation of the nitrogen nutrient status was explored. The tomato variety “FENOUYA” was used as the test crop, and aerated drip irrigation was adopted by setting three levels of aeration rates, namely, A1 (dissolved oxygen concentration of irrigation water is 5 mg L^−1^), A2 (dissolved oxygen concentration of irrigation water is 15 mg L^−1^), and A3 (dissolved oxygen concentration of irrigation water is 40 mg L^−1^), and three levels of nitrogen rates, namely, N1 (120 kg ha^−1^), N2 (180 kg ha^−1^) and N3 (240 kg ha^−1^). The model of the critical nitrogen concentration dilution of tomatoes under different aerated treatments was established. The results showed that (1) the dry matter accumulation of tomatoes increased with the increase in the nitrogen application rate in a certain range and it showed a trend of first increase and then decrease with the increase in aeration rate. (2) As the reproductive period progressed, the nitrogen concentration in tomato plants showed a decreasing trend. (3) There was a power exponential relationship between the critical nitrogen concentration of tomato plant growth and above-ground biomass under different levels of aeration and nitrogen application rate, but the power exponential curves were characterized by A1 (*Nc* = 15.674*DM*^−0.658^), A2 (*Nc* = 101.116*DM*^−0.455^), A3 (*Nc* = 119.527*DM*^−0.535^), N1 (*Nc* = 33.819*DM*^−0.153^), N2 (*Nc* = 127.759*DM*^−0.555^) and N3 (*Nc* = 209.696*DM*^−0.683^). The standardized root mean square error (n-RMSE) values were 0.08%, 3.68%, 3.79% 0.50%, 1.08%, and 0.55%, which were less than 10%, and the model has good stability. (4) The effect of an increased nitrogen application rate on the critical nitrogen concentration dilution curve was more significant than that of the increase in aeration rate. (5) A nitrogen nutrition index model was built based on the critical nitrogen concentration model to evaluate the nitrogen nutritional status of tomatoes, whereby 180 kg ha^−1^ was the optimal nitrogen application rate, and 15 mg L^−1^ dissolved oxygen of irrigation water was the optimal aeration rate for tomatoes.

## 1. Introduction

As a major element for plant growth, nitrogen (N) plays a vital role in agricultural production [1]. Currently, in order to maximize crop yields, excess N fertilizer is applied well beyond the optimum levels required for crops [2]. This leads to a serious waste of resources, and even causes more serious environmental problems [3], such as water pollution [3], soil pollution [4], and air pollution [5] associated with nitrogen fertilizers, which have become pressing issues in agricultural research today. Therefore, rational nitrogen fertilizer management is a prerequisite for crops to achieve the target yields [6], and the rational application of nitrogen fertilizer on the premise of ensuring high yields has become an urgent problem in crop production. Investigations have revealed a lack of accuracy in extrapolating the tomato N uptake status due to the amount of N applied and the amount of change in soil N [7]. The dynamics in N concentration throughout the life span of tomatoes can more accurately reflect the nitrogen nutritional status of the crop in each period, and thus identify the soil nitrogen supply capacity and the nitrogen uptake capacity of the tomato crop at each stage [8]. As early as 1952, Ulrich [9] introduced the concept of the “critical nitrogen concentration”, that is, the minimum nitrogen concentration when the maximum biomass is reached during a certain growth phase [10], which is the basic method for nitrogen diagnosis in tomatoes. Previous studies have found that the plant nitrogen concentration tends to decrease with the accumulation of the aboveground dry matter biomass of the crop. By constructing a relationship between the critical nitrogen concentration and aboveground dry matter biomass, the value of the critical plant nitrogen concentration can be quickly calculated.

The accurate diagnosis of a plant’s nitrogen nutritional status is the basis for rational fertilizer application. Much research has been conducted on spectral remote sensing [11], satellite data [12], and machine vision [13] for the accurate diagnosis of plant nitrogen nutrition. However, the cost of these technologies makes it difficult for them to be popularized. Many critical N concentration dilution curve models have been established for pepper [14], wheat [15], maize [16], cotton [17], winter oilseed rape [18], and tomatoes [19]. Nitrogen-deficit accumulation modeling based on critical nitrogen concentration curves is an effective method for the nitrogen nutritional diagnosis of crops [16,20]. Currently, most of the research on critical N concentration modeling focuses on the status of nitrogen gain or loss and utilization in facility crops under different water and fertilizer conditions [21]. However, the dilution curve of the N concentration in tomatoes under aerated drip irrigation in greenhouses with controlled-release N fertilizer has not been clarified so far. Soil hypoxia stress is one of the main manifestations of soil degradation in facility agriculture. When soil aeration is insufficient or the oxygen content is low, the plant root system’s ability to absorb water and nutrients from the soil significantly decreases [22], affecting the tomato dry matter biomass, yield, and nitrogen uptake [23]. Aerated drip irrigation delivers oxygen-enriched irrigation water to the crop root zone through an irrigation pipeline system, which can be an effective solution to the problem of hypoxia in the rhizosphere [24]. Therefore, by studying the dynamics in nitrogen concentration in tomato plants at different reproductive stages under aerated drip irrigation, nitrogen concentration curves were established to increase the tomato yields and nitrogen use efficiency, while reducing the rate of nitrogen applied. Therefore, in this paper, tomato was used as the test crop, and two-factor, three-level randomized block trials were set up for the aeration (A1: 5 mg L^−1^; A2: 15 mg L^−1^; A3: 40 mg L^−1^) and nitrogen application rates (N1: 120 kg ha^−1^; N2: 180 kg ha^−1^; and N3: 240 kg ha^−1^). The aim was to investigate the adaptability of the critical nitrogen concentration dilution curve under aerated irrigation, providing a theoretical basis and technical support for precise irrigation and nitrogen addition to tomatoes under fertigation.

## 2. Results

### 2.1. Changes in Aboveground Biomass (Stems, Leaves, and Fruits) and Nitrogen Uptake in Tomato under Different Aeration/Nitrogen Combinations

#### 2.1.1. Above-Ground Dry Matter Accumulation in Tomato

Figure 1 shows the dynamic accumulation process of the above-ground biomass of greenhouse tomatoes under different levels of aeration and N application. The total dry matter mass of tomato increased with increasing nitrogen applications, and the accumulation of dry matter of tomato in the aerated treatments was significantly higher than that of the control treatment. At flowering and fruiting stage, there was no significant difference (*p* > 0.05) in dry matter accumulation of tomatoes among treatments. At the fruit expansion stage, the overall accumulation of dry matter in tomatoes in A2 treatment significantly increased by 29.9% and 15.5% compared to that in A1 and A3 treatments (*p* < 0.05), respectively. At the harvest stage, the overall accumulation of dry matter in tomatoes in the A2 treatment was significantly higher than that in A1 and A3 treatments (*p* < 0.05), by 25.2% and 20.3% overall, respectively.

The amount of fertilizer application promoted fruit dry matter accumulation in all cases with the same amount of aerated air. Tomato dry matter mass increased with the increase in fertilizer application at the fruit expansion stage and ripening stage, whereas different aeration and nitrogen application rates at flowering and fruiting did not significantly (*p* > 0.05) affect tomato dry matter mass. At the flowering and fruiting stage, the overall accumulation of dry matter in tomato in N2 and N3 treatments was significantly higher than that in N (*p* < 0.05). At the fruit expansion stage, the overall accumulation of dry matter in tomatoes in the N3 treatment was significantly higher than that in N1 and N2 treatments, by 23.9% and 19.2% overall, respectively. At harvest stage, the overall accumulation of dry matter in tomatoes in the N3 treatment was significantly higher than that in N1 and N3 treatments, by 25.1% and 13.3% overall, respectively. Among them, the A2N3 treatment was the most favorable for tomato dry matter accumulation, followed by the A2N2 and A3N3 treatments.

#### 2.1.2. Tomato Nitrogen Uptake and Accumulation

Dynamics in nitrogen uptake by tomatoes under aerated conditions are shown in Figure 2. The results showed that the cumulative amount of nitrogen increased with the elapsed days of tomato planting, and the cumulative amount of nitrogen uptake of tomato organs at different fertility stages as a whole were as follows: fruit > leaf > stem. When the aerated amount was the same, the cumulative uptake of tomato nitrogen increased with the increase in nitrogen application, and when the nitrogen application was the same, the cumulative uptake of tomato nitrogen was shown to be in descending order: A2 > A3 > A1.

At the flowering and fruiting stages, nitrogen uptake by the plant was low, and with the increase in nitrogen application, tomato nitrogen uptake increased, but there was no significant difference in nitrogen uptake under different aerated treatments (*p* > 0.05). The accumulation of the nitrogen uptake by the tomatoes showed a rising trend at the fruit expansion stage, accounting for 32.2–35.4% of total N uptake, and the nitrogen uptake in A2 was significantly higher than that of A1 and A3. The nitrogen allocation proportion in fruit was the largest, which accounted for 47.6–57.8% of the plant. The nitrogen accumulation of the plant reached 394.5–794.1 kg ha^−1^ at the harvest stage. Among them, the cumulative nitrogen uptake in the A2N3 treatment increased by 20.3% and 25.2%, compared with that in the A3N3 treatment and A1N3, respectively (*p* < 0.05). Nitrogen allocation in fruit accounted for 50.5–60.1%, and there was an increasing trend of nitrogen allocation in tomato leaves.

#### 2.1.3. Analysis of the Influence of Aerated Drip Irrigation on Tomato Yield

The effect of different aeration and nitrogen application rates on tomato yield is shown in Figure 3. Compared with the non-aerated irrigation, the yields of aerated irrigation increased by 13.35–29.73%, respectively. The highest yield of A2N3 treatment was 250.42 t ha^−1^, which was 29.73% higher than that of A1N3 treatment. The yield of tomato also increased significantly with the same gas addition (except for A1N2 and A1N3). When the aeration rate was A2, the yield of the N3 treatment was 8.46% and 37.86% higher than that of theN2 and N1 treatments, respectively. When the aeration rate was A3, the yield of the N3 treatment increased by 13.24% and 28.25% compared with that of the N2 and N1 treatment, respectively. Under the same fertilizer application rate, the yield of A2N3 treatment was 11.53% and 29.73% higher than that of the A3N3 and A1N3 treatment, respectively, and the yield of the A2N2 treatment increased by 16.45% and 22.81% compared with that of the A3N2 and A1N2 treatments, respectively. Both nitrogen application and aeration rate significantly increased tomato yield, and the increase in tomato yield was significant (*p* < 0.05) at the aeration rate of A2.

### 2.2. Modeling and Testing of Critical Nitrogen Concentration Dilution Curves

#### 2.2.1. Modeling of Critical Nitrogen Concentration Dilution Curves for Tomatoes

The critical nitrogen concentration dilution curve model of tomato plants under three aeration and three nitrogen application conditions was established. When the biomass was less than 1.0 t ha^−1^, there was no significant decreasing trend in the aboveground nitrogen concentration values of facility cultivation of tomato plants with the increase in biomass, so the nitrogen concentration values in this stage were not simulated. When the above-ground biomass was greater than 1.0 t ha^−1^, plant nitrogen concentration and the corresponding above-ground biomass data were fitted according to Equation (2). A model of the critical nitrogen concentration dilution curves of tomato plants under the three aeration and three nitrogen application was established, and two nitrogen dilution boundary models were obtained by using the maximum (*Nmax*) and minimum (*Nmin*) nitrogen concentration on each sampling day. The determination coefficients R^2^ of the A1, A2, and A3 models were 0.921, 0.943, and 0.858, respectively; those of the N1, N2, and N3 models were 0.815, 0.954, and 0.934, reaching the significant level, which indicates that the model could well explain the relationship between critical nitrogen concentration and above-ground dry matter accumulation in aerated drip irrigation tomatoes (Figure 4). Under the same aboveground biomass, the nitrogen concentrations were highly variable, and the maximum (*Nmax*) and minimum (*Nmin*) nitrogen concentration on each sampling day could be fitted for the maximum nitrogen dilution model and the minimum nitrogen dilution model; the results were also consistent with model (2).

With the increase in aeration rate, the slope of the critical nitrogen concentration dilution curve decreased and then increased (i.e., the value showed an overall trend of decreasing and then increasing). Meanwhile, with the increase in nitrogen application, the slope of the critical nitrogen concentration dilution curve increased (i.e., the value was increasing). The effect of the increasing nitrogen rates on the critical nitrogen concentration dilution curve was more significant than that of increasing aeration rates (Figure 4).

#### 2.2.2. Model Validation of Critical Nitrogen Concentration Dilution Curves for Tomatoes

The above model was validated using the experimentally measured data, and as seen in Figure 5, the fit between the simulated values and the real values reached a highly significant degree, and the two basically conformed to the linear relationship of *y* = *x*, indicating a large degree of conformity. Among them, the *RMSE* under the A1, A2, A3, N1, N2, and N3 treatments was 0.02 g kg^−1^, 0.82 g kg^−1^, 0.92 g kg^−1^, 0.11 g kg^−1^, 0.27 g kg^−1^, and 0.16 g kg^−1^, respectively. The standardized root mean square error (*n-RMSE*) was 0.08%, 3.68%, 3.79%, 0.50%, 1.08%, and 0.55%, respectively, which were less than 10%, indicating that the model had good stability and could be further used to estimate tomato nitrogen nutrition.

### 2.3. Modeling of Nitrogen Nutrient Indices in Aerated Tomatoes

The dynamics of the nitrogen nutrition index (*NNI*) of tomato plants are shown in Figure 6. The *NNI* under each aeration treatment increased with the increase in nitrogen application, and generally showed a decreasing trend with the elapse of sampling time. The *NNIs* of N1 and N2 under A1 were less than 1, indicating that the nitrogen supply was insufficient at this time and the tomato plants were short of nitrogen. Meanwhile, the *NNIs* of N3 were greater than 1, indicating that the amount of nitrogen applied was too much and the tomato plants were over-nourished. Under the A2 treatment, the *NNIs* of N1 were all less than 1, indicating that the nitrogen supply was insufficient at this time, and the tomato plants were nitrogen-deficient; the *NNIs* under the N2 treatment were close to 1, indicating that this nitrogen application was the optimal level; the *NNI* values under the N3 treatment were greater than 1, indicating that this nitrogen application was too much, and the tomato plants were over-nourished. Under the A3 treatment, the *NNIs* of N1 were all less than 1, indicating insufficient nitrogen supply and nitrogen shortage in the tomato plants; the *NNIs* of the N2 treatment were close to 1, indicating that this nitrogen application was the optimal level; and the *NNIs* of the N3 treatment were greater than 1, indicating that this nitrogen application was too much and the tomato plants were over-nourished. Therefore, it was concluded that the appropriate amount of nitrogen application in the growth stage under the different aerated treatments were A1: (N3, 240 kg ha^−1^), A2: (N2, 180 kg ha^−1^) and A3: (N2, 180 kg ha^−1^).

As can be seen from Figure 6, the tomatoes were over-nourished under the aeration rate of A2 due to a *NNI* > 1. The *NNIs* of A1 and A3 were both lower than 1, which was unfavorable for the growth and development of tomato plants.

### 2.4. Factor Analysis

Factor analyses was performed based on tomato aboveground dry matter accumulation, tomato nitrogen uptake accumulation, tomato yield, aeration, and nitrogen application. A principal component was extracted based on eigenvalues greater than 1 and its cumulative variance contribution was 96.725% (Table 1). As seen from Table 1, the overall rankings at the level of aerating are A2 > A3 > A1, and at the level of nitrogen application, they are N2 > N3 > N1. Their overall rankings are A2N2 > A2N3 > A3N2 > A3N3 > A1N2 > A1N3 > A2N1 > A3N1 > A1N1.

## 3. Discussion

### 3.1. Characteristics of Critical Nitrogen Concentration Dilution Curve Model of Tomatoes under Aeration Conditions

Nitrogen is the macronutrient for tomatoes, and it is involved in most of the physiological metabolic processes. Reasonable application of nitrogen fertilizer is the key to regulate the growth and development of tomatoes, improve the function of leaf blades, and increase the yield [25,26,27]. However, irrational application of nitrogen fertilizer and low utilization efficiency in the current production is very common, so it is very important to clarify the characteristics of tomato nitrogen demand for reasonable fertilizer application [28]. In studies on critical N concentrations of aboveground dry matter in wheat, oilseed rape, potato, and sorghum [29,30,31,32,33], the critical N concentration model is suitable for different climatic regions and crop varieties. A critical nitrogen dilution model of facility tomatoes at different aeration levels and nitrogen application rates was constructed in this study: A1 (*Nc* = 15.674*DM*^−0.658^), A2 (*Nc* = 101.116*DM*^−0.455^), A3 (*Nc* = 119.527*DM*^−0.535^), N1 (*Nc* = 33.819*DM*^−0.153^), N2 (*Nc* = 127.759*DM*^−0.555^), and N3 (*Nc* = 209.696*DM*^−0.683^). This was consistent with the previous models of critical nitrogen dilution for different crops, but with different parameters. This study showed that the parameter value *b* under the A1 treatment was the largest, suggesting that *DM*^−*b*^ was the smallest one, and the final total dry matter accumulation of tomato plants under the A1 treatment was correspondingly the smallest. The parameter value *b* under A2 treatment was the smallest, which meaned that *DM*^−*b*^ was the largest, indicating that the moderate aerated amount could promote nitrogen uptake and improve the nitrogen holding capacity of tomato plants. However, the dry matter accumulation of tomato plants under the same batch of sampling was still smaller than that of the A3 treatment, whereas the maximum theoretical aboveground biomass was larger than that of the A3 treatment, indicating that the A3 treatment reduced the rate of dry matter accumulation of tomato, but increased the total amount of final dry matter accumulation of tomato plants, which might result in the prolongation of the fertility period of tomato plants [34]. For the critical nitrogen concentration dilution curves at different levels of nitrogen application (N1, N2, and N3), the slopes of the critical nitrogen concentration dilution curves increased as a whole as the amount of nitrogen applied increased (Figure 4). This indicated that an increase in nitrogen fertilizer (within certain limits) promoted the ability of the tomato crop to absorb nitrogen from the soil. 

Although the critical nitrogen concentration dilution curve model based on the experimental data could well describe the relationship between above-ground biomass and nitrogen concentration, only one variety was selected. It is still necessary to test the model using experimental information from other ecological sites and species that are independent of the modeling data. In the future, it is necessary to further introduce different ecological locations, tomato varieties, and use different cultivation densities and nitrogen fertilizer transport methods to verify whether the above factors will have an impact on the parameters of the model of the critical nitrogen concentration dilution curve of processed tomatoes, further improving the generalizability of the model.

### 3.2. Nitrogen Nutrition Diagnosis and Recommended Nitrogen Fertilizer and Aeration Application Rates for Tomatoes

Based on the dilution curve of critical nitrogen concentration, a nitrogen nutrient index model was established using the ratio of actual nitrogen concentration to critical nitrogen concentration. It could accurately diagnose and measure the nitrogen nutritional status of the plant at the reproductive stage, which is of great significance for the quantification of the amount of fertilizer applied during crop growth and development [35]. *NNI* visually reflects the nitrogen status of the crop, and it increases with the increase of nitrogen application. This suggests that increasing the amount of nitrogen could increase nitrogen uptake and accumulation. *NNI* = 1 indicates that the crop was in a suitable nitrogen status [36,37]. According to the relationship between *NNI* and the horizontal line “1” at different levels of aeration (Figure 6), the *NNI* of the A1 treatment ranged from 0.789 to 1.237, that of A2 treatment was between 0.828 and 1.258, and that of A3 treatment ranged from 0.772 to 1.242. The *NNI* of N2 fluctuated around “1” at each aerated level (A1, A2, and A3) because dry matter accumulation did not increase significantly after nitrogen application reached the N2 level. The critical N concentration obtained from the maximum dry matter accumulation simulation was closer to the aboveground biological N concentration at the N2 level, so the *NNI* of each aeration level was closer to “1” when nitrogen application reached the N2 level. The relationship between *NNI* and above-ground dry matter accumulation and yield was further explored. When *NNI* was less than a certain value, the dry matter production in the aboveground part increased with the increase in *NNI*, and when *NNI* reached a certain value, the production entered a plateau stage and no longer increased significantly, and the value of this value ranged from 0.978 to 1.038, which is close to 1, and this value range was the same as the *NNI* range of N2. Therefore, the recommended N application rate was initially determined as N2 (180 kg ha^−1^). For the different nitrogen application treatments (N1, N2, and N3), the *NNI* was greater than 1 at the aeration rate of A2, indicating that the tomato plants were over-nourished (Figure 6). When the aeration rate reached the A3 level (much higher than A2), it was not conducive to the growth and development of tomato plant [38,39]. Therefore, the recommended aeration rate was initially determined as A2. The results are consistent with the comprehensive evaluation of factor analysis (Table 1), in which the A2 treatment was the optimal aeration rate and the N2 treatment was the optimal nitrogen application rate, i.e., the A2N2 treatment was optimal.

## 4. Materials and Methods

### 4.1. Site Description and Experimental Design

This experiment was conducted from 9 March to 10 July 2019 in a solar greenhouse of the Experimental Farm of the Agricultural Efficient Water Use in North China University of Water Resources and Hydropower, Zhengzhou City, Henan Province, China (34°47′5.91″ N, 113°47′20.15″ E). The solar greenhouse has a ridge type with a span of 9.6 m and an opening of 4 m, with a total area of 537.6 m^2^. The test soil was a loamy clay loam, with a pH of 7.50, soil capacity of 1.25 g cm^−3^, and maximum field water holding capacity in the field of 36.64%. The mass fractions of sandy grains (>0.02–2 mm), chalky grains (0.002–0.02 mm), and clayey grains (<0.002 mm) were 20.95%, 36.51%, and 42.54%, respectively. The soil organic carbon, alkali-hydrolyzable nitrogen, available phosphorus, and available potassium contents before planting were 19.38 g kg^−1^, 38.87 g kg^−1^, 8.68 mg kg^−1^, and 4.58 mg kg^−1^, respectively.

### 4.2. Experimental Design and Management

The tomato variety tested was “FENOUYA” for its resistance to virus diseases and nematodes. This experiment was conducted in buckets, which were cylindrical, 30 cm in diameter and 40 cm in height, with a drip head (Netafim, NETAFIM (Beijing) Agricultural Technology Co., Ltd, Beijing, China) buried in the center of each bucket at a depth of 15 cm, with a first irrigation pressure of 0.1 MPa and a drip head flow rate of 2.7 L h^−1^. Tomato seedlings were transplanted when they were in the 3-leaf-1-heart to 4-leaf-1-heart range, 1 plant per pot. The planting buckets were fully buried into soil. We watered thoroughly on the day of transplanting and covered the soil surface with silver and black plastic film on the 15th day after transplanting, thus keeping it moist and weed-free. When a plant’s height was 30–40 cm, it was hung and topped after three spikes of fruits. The whole growth period was 124 days, and its division is shown in Table 2.

The design of 2-factor, 3-level randomized block groups for aeration and nitrogen application, with a total of 9 experimental treatments and 10 replications for each treatment, is shown in Table 3.

As shown in Figure 7, aerated drip irrigation was carried out using a venturi injector (Mazzei air injector 684, Mazzei Injector Corp, Bakersfield, CA, USA) and Aquamarine Technology micro- and nano-bubbler (50 HZ, Yixing Aquamarine Technology Co., Ltd., Yixing, China), respectively. Among them, water storage lines, circulating water pumps, a venturi air injector, and other equipment were used to adsorb air for aeration through pressure differences, and irrigation began when the dissolved oxygen of the irrigation water reached 15 mg L^−1^. The Aquamarine Technology Micro-nano Bubble generator adopts the principle of pressure swing adsorption separation to prepare pure oxygen, which is circulated through the external water storage tank to prepare ultra-highly dissolved oxygen in the micro-nano bubbler water. In the test, the main water supply pipe, pressure gauge, and water supply meter were connected at the end of the circulating aeration outlet. Irrigation started when the dissolved oxygen of the irrigation water reached 40 mg L^−1^. The water supply pressure of the water supply device was controlled at 0.10 MPa.

From 08:00 to 09:00 a.m. every day, the evaporation of a Φ601 standard evaporation dish and moisture probe readings were measured to control the amount of irrigation water, which was calculated according to the following Formula (1).
(1)$${W=A\cdot E}_{P}{\cdot K}_{P}$$
where *W* is the amount of water per irrigation for each treatment, L; *A* is the area of the potting bucket, m^2^; *E_P_* is the evaporation from the evaporating dish at the interval of 2 irrigations, mm; and *K_P_* is the coefficient of the evaporating dish, which is 1.0.

This experiment adopted water–fertilizer–air-coupled fertigation with equal amounts of phosphate and potassium at each treatment. Nitrogen fertilizer was applied separately during the whole growth period, on the 10th, 25th, 46th, 60th, and 74th d after planting, with the nitrogen application ratio of 1:1:2:2:2. The fertilizer scheme for greenhouse tomatoes is shown in Table 4.

### 4.3. Test Indicators and Methods

#### 4.3.1. Above-Ground Plant Biomass

A total of three destructive samples were taken at the tomato flowering and fruiting stage, fruit expansion stage, and harvesting stage (34th, 81th, and 124th d after transplantation). Three tomato plants of uniform growth were selected for each treatment in pots. The three parts of tomato, stem, and leaf and fruit were placed in a drying oven at 105 °C for 30 min and then adjusted to 75 °C to be dried to constant weight and the dry mass was weighed. 

#### 4.3.2. Nitrogen Content of Above-Ground Plants

The dry samples from each organ treatment were crushed, ground, and passed through a 0.25 mm sieve. The total nitrogen content of each part of the tomato plant was measured using a Kjeldahl nitrogen meter (K9840, China Haineng Instrument Co., Ltd., Jinan, China), and finally the aboveground nitrogen concentration of the tomato plants was calculated.

#### 4.3.3. Yield

Three tomato plants of uniform growth were selected for each treatment, which were picked and pulled after ripening. The number of fruits and single fruit weight of each plant were determined separately and used to calculate tomato yield.

### 4.4. Modeling

#### 4.4.1. Modeling of Critical Nitrogen Dilution Curves

Based on the method of calculating the critical nitrogen concentration proposed by Justes et al. [40] and the method of constructing the critical nitrogen concentration dilution curve model of Makowski et al. [41], the steps were briefly described as follows: (1) determination of the aboveground biomass and its corresponding nitrogen concentration value of each sample. (2) An analysis of ANOVA variance was performed on the aboveground biomass of the crop, and the data were divided into nitrogen nutrient-constrained and non-constrained groups according to whether the crop growth was constrained by nitrogen nutrition. (3) The relationship between the aboveground biomass and its nitrogen concentration value of the nitrogen nutrient-constrained group was fitted using a linear curve, and the average value of the aboveground biomass of the non-constrained group was taken as a representative of the maximum value of the biomass. The theoretical critical nitrogen concentration for each sample was determined by the intersection of the vertical lines in the above curve where the maximum biomass was the abscissa (i.e., the corresponding ordinate). According to the definition of the critical nitrogen concentration proposed by Lemaire et al. [42], the modeling expression is listed as follows:(2)$${N_{c}=a\cdot W}^{-b}$$
where *N_c_* is the value of critical nitrogen concentration, g kg^−1^; *a* is the critical nitrogen concentration of the plant in the above-ground biomass, 1.00 t ha^−1^; *W* is the accumulated aboveground dry matter biomass, t ha^−1^; *b* is the statistical parameter for the slope of the critical nitrogen concentration dilution curve.

#### 4.4.2. Verification Model

A regression standard error of estimate (*RMSE*) was used to analyze the compliance of the simulated values with the real values. The smaller the *RMSE* value, the better the simulated values fit and the smaller the deviation, the higher the prediction accuracy of the mode. At the same time, a 1:1 histogram between the simulated values and the real values was used to visualize the fit degree and reliability of the model [16].
(3)$$RMSE=\sqrt{\frac{\sum_{i=1}^{n}(P_{i}-O_{i})²}{n}}$$
where *P_i_* is the measured value, *O_i_* is the corresponding simulated value, and *n* is the sample size.

#### 4.4.3. Nitrogen Nutrition Index (*NNI*) Modeling

To further clarify the nitrogen nutritional status of crops, Lemaire et al. proposed the concept of a nitrogen nutritional index [43], which was listed as
(4)$$NNI=\frac{N_{t}}{N_{c}}$$

In the formula, *N_t_* is the measured value of the aboveground nitrogen concentration of tomatoes, and *N_c_* is the value of critical nitrogen concentration according to the model of critical nitrogen concentration dilution. *NNI* = 1 indicates that the nitrogen nutrient level of the plant is in the optimal state, *NNI* > 1 shows that the nitrogen nutrient is in excess, and *NNI* < 1 indicates that the nitrogen nutrient is insufficient.

### 4.5. Statistical Analysis and Graphing

Excel 2019 was used for data processing. Duncan’s new complex polar deviation method was used for significance testing through SPSS 22.0 software and the test level was 0.05. A factor analysis was performed using SPSS 22.0 software. The plotting was performed through Origin 2021 software.

## 5. Conclusions

Within a certain range, tomato dry matter accumulation increased with the increase in nitrogen application rate, and it showed a tendency of first increase and then decrease with the increase in aeration rate. However, as the growth period progressed, the nitrogen concentration of tomato plants showed a decreasing trend. Compared with the increase in aeration rate, the increase in nitrogen application rate had more significant influence on the dilution curve of critical nitrogen concentration. In addition, there was a power exponential relationship between critical nitrogen concentration and above-ground biomass of tomato plants., and its standardized root-mean-square errors were 0.08%, 3.68%, 3.79%, 0.50%, 1.08%, and 0.55%, which were all less than 10%. The model had excellent stability and small error range. The nitrogen nutrition index model was constructed based on the critical nitrogen concentration model to measure the nitrogen nutritional status during the growth and development of tomatoes under aerated irrigation, i.e., 180 kg ha^−1^ was the optimal nitrogen application rate, and 15 mg L^−1^ was the optimal aeration rate.

Further introduction of crops from different ecological sites and tomato varieties with different management practices such as water addition, nitrogen application, and soil aeration are needed in the future to further improve the generalizability of the model.

## Figures and Tables

**Figure 1 plants-13-00270-f001:**
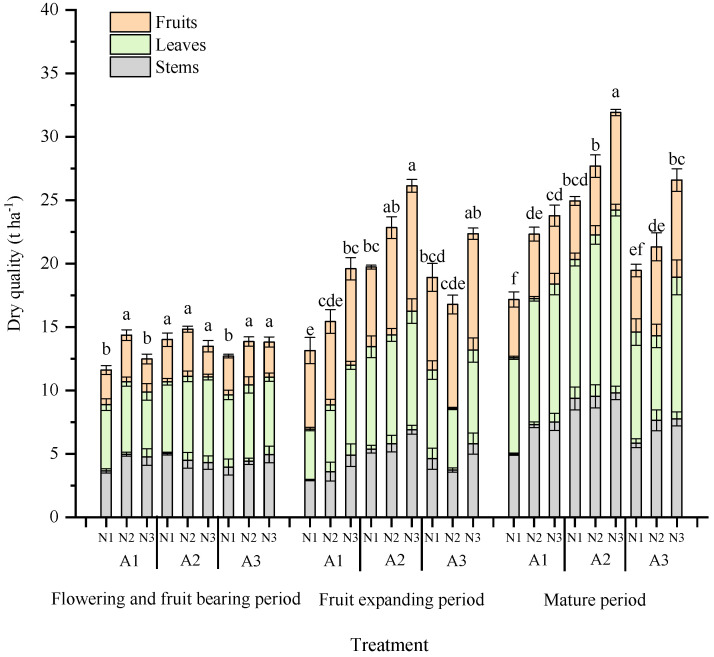
Effects of nitrogen fertilizer on aboveground biomass of greenhouse tomatoes at different aeration rates. Different lower-case letters on the bars indicate significant differences between treatments in each period (*p* < 0.05). N1, N2, and N3 are 3 nitrogen levels of 120, 180, and 240 kg ha^−1^, respectively. A1, A2, and A3 are 3 aeration rates of 5, 15, 40 mg L^−1^, respectively.

**Figure 2 plants-13-00270-f002:**
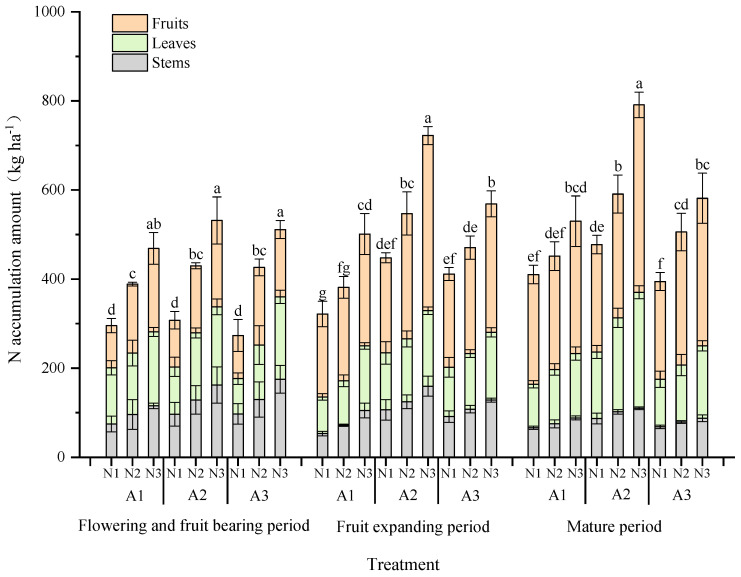
Nitrogen accumulation dynamics in the aboveground part of tomato under different aeration rates. Different lower-case letters on the bars indicate significant differences between treatments in each period (*p* < 0.05).

**Figure 3 plants-13-00270-f003:**
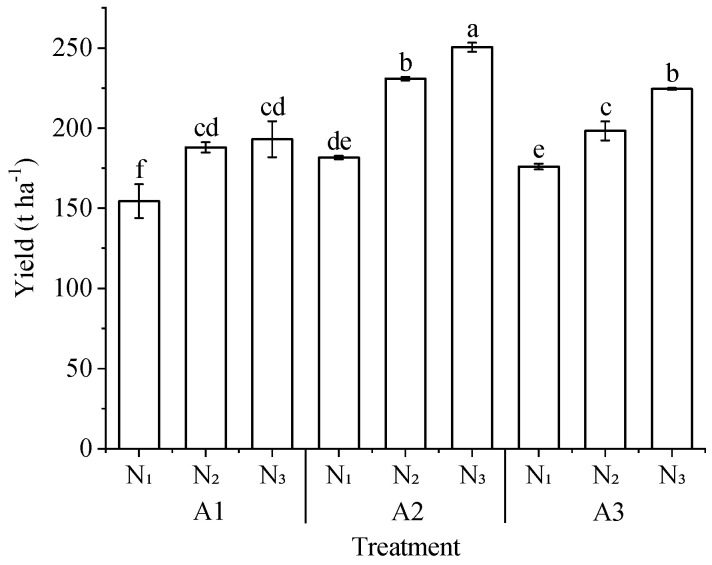
Tomato yield under aerated drip irrigation. Different lower-case letters (a–f) on the bars indicate significant differences among the treatments (*p* < 0.05).

**Figure 4 plants-13-00270-f004:**
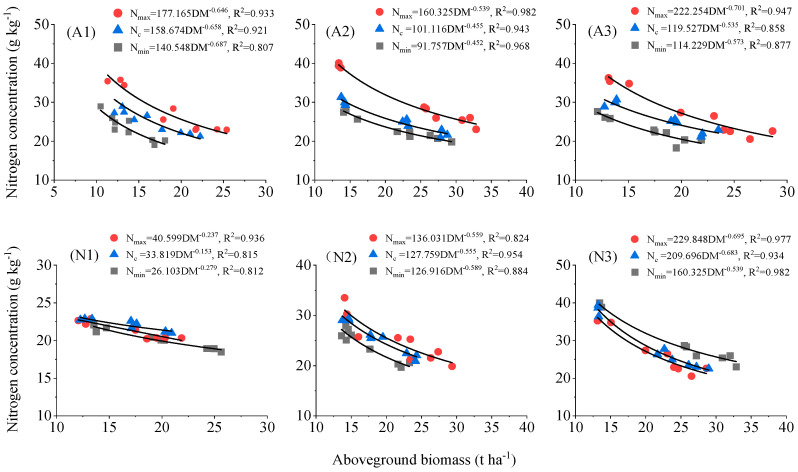
Dilution curves of nitrogen concentration in tomato above-ground biomass under different aeration conditions.

**Figure 5 plants-13-00270-f005:**
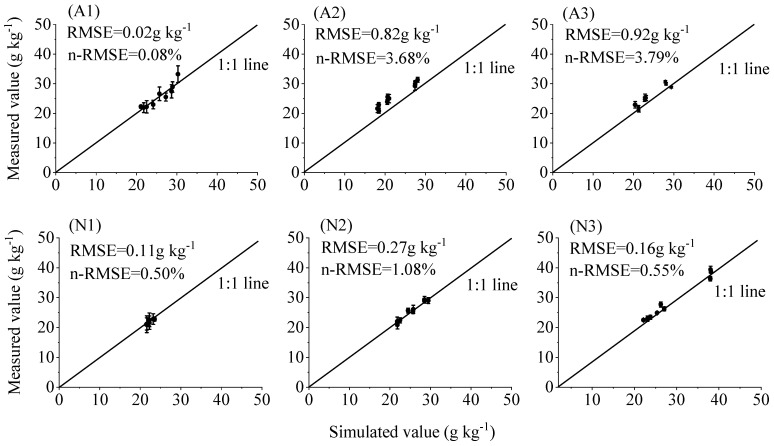
Comparison of simulated and observed values of critical nitrogen content in tomato plants under different aeration conditions.

**Figure 6 plants-13-00270-f006:**
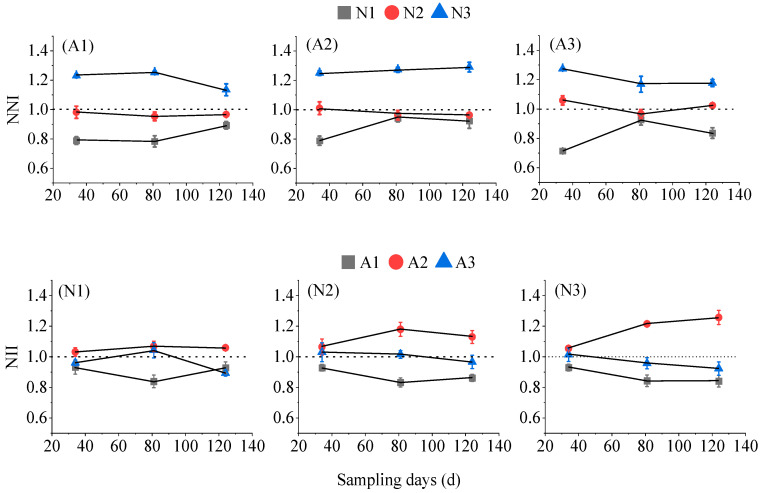
The process of nitrogen nutrition index (*NNI*) changes in tomato plants under different aerated conditions.

**Figure 7 plants-13-00270-f007:**
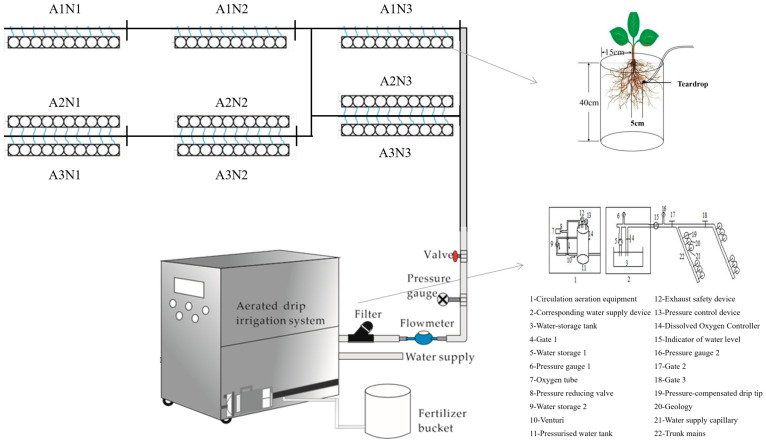
Schematic layout of tomato pot planting area and aerated irrigation system.

**Table 1 plants-13-00270-t001:** Comprehensive evaluation of the tomato growth based on factor analysis.

Treatment	Overall Score	Overall Ranking
A1N1	−1.19563	9
A1N2	−0.19575	5
A1N3	−0.50275	6
A2N1	−0.53809	7
A2N2	2.03413	1
A2N3	0.86384	2
A3N1	−0.95248	8
A3N2	0.44372	3
A3N3	0.04301	4

Note: N1, N2, and N3 are low-, medium-, and high-nitrogen treatments, respectively; A1, A2, and A3 are the control, medium-, and high-aeration treatments, respectively.

**Table 2 plants-13-00270-t002:** Duration of tomato growth period.

Growth Period	Start Date	End Date	Days after Transplanting
Seeding period	9 March 2019	2 April 2019	1–25
Flowering and fruit bearing period	3 April 2019	17 April 2019	26–40
Fruit expanding period	18 April 2019	21 May 2019	41–74
Mature period	22 May 2019	10 June 2019	75–124

**Table 3 plants-13-00270-t003:** Experimental design.

Treatment	Nitrogen Application Rate (kg ha^−1^)	Aeration Rate (mg L^−1^)
A1N1	120	5
A1N2	180	5
A1N3	240	5
A2N1	120	15
A2N2	180	15
A2N3	240	15
A3N1	120	40
A3N2	180	40
A3N3	240	40

Note: N1, N2, and N3 are low-, medium-, and high-nitrogen treatments, respectively; A1, A2, and A3 are the control, medium-, and high-aeration treatments, respectively.

**Table 4 plants-13-00270-t004:** Fertilizer application rate during tomato cropping.

Days after Transplanting	N (kg ha^−1^)			P_2_O_5_ (kg ha^−1^)	K_2_O (kg ha^−1^)
	N_1_	N_2_	N_3_		
10	15	22.5	30	24	30
25	15	22.5	30	24	30
46	30	45	60	24	30
60	30	45	60	24	30
74	30	45	60	24	30
Total	120	180	240	120	150

## Data Availability

Data are contained within the article.

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
