# Peer review of "Comprehensive Evaluation of Tomato Growth Status under Aerated Drip Irrigation Based on Critical Nitrogen Concentration and Nitrogen Nutrient Diagnosis"

_plants, 2024, doi:10.3390/plants13020270_

Round 1
Reviewer 1 Report
Comments and Suggestions for Authors
The manuscript is well organized and easy to red. The novelty is important and figures and tables well presented. I have just few comments:
Line 106 exchangeable potassium?
Please report the values in hectare that hm2
Why fenoya cultivar was selected?
Why was the experiment performed in a pot?
Discussion should be improved.
The limitation of the study should be reported in the conclusions.
Author Response
For research article
Response to Reviewer 6 Comments
|
||
1. Summary |
|
|
Thank you very much for taking the time to review this manuscript. According to your comments, the manuscript is revised in yellow. Revision notes, point-to-point, are given as follows: |
||
2. Point-by-point response to Comments and Suggestions for Authors |
||
Comments 1: Line 106 exchangeable potassium? |
||
Response 1: Soil available potassium concentrations were analyzed with the ammonium acetate (NH4OAc) extraction method. (Song, X.-D., Liu, F., Wu, H.-Y., Cao, Q., Zhong, C., Yang, J.-L., Li, D.-C., Zhao, Y.-G., and Zhang, G.-L., Effects of long-term K fertilization on soil available potassium in East China. CATENA, 2020. 188: p. 104412.) |
||
Comments 2: Please report the values in hectare that hm2 Response 2: Thanks to reviewer for reminder, we have revised lines 19, 20, 35, 87, 121, 151, 189, 190, 246, 264, 276, 293, 296, 357, 444, 451, 466 in the manuscript.
Comments 3: Why “fenouya” cultivar was selected? Response 3: The tomato variety "fenouya" has the advantages of good resistance to chemicals, high yields, large fruits, and is effective in aeration and irrigation. Comments 4: Why was the experiment performed in a pot? Response 4: Due to the relatively small size of the pots, their conditions such as fertiliser, water, temperature, humidity and light are easy to control. This creates favourable conditions for ensuring the success and precision of experiments.
Comments 5: Discussion should be improved. Response 5: A "factor analysis" has been added to the results and discussion to better understand the effects of aeration and nitrogen application and the relationship between them. We have added lines 365-375, 449-452 in the manuscript.
Comments 6: The limitation of the study should be reported in the conclusions. Response 6: The conditions of pot experiments and field experiments are different, so after effective results are obtained, field experiments must be conducted to verify them before they can be promoted in production. We have added lines 468-470 in the manuscript.
3. Additional clarifications |
The description of the test method was supplemented, we have revised lines 156-159, 162-164, 213 in the manuscript.

Reviewer 2 Report
Comments and Suggestions for Authors
In my opinion is an interesting work, with scientific and applied value. Some things that I think the paper need are;
1. Much more care in the manuscript writing. I have highlighted in red typos words and phrases whose writing is wrong or meaning is difficult to understood. These should be corrected. I think there are more to be corrected.
2. Adding a factorial analysis to the data will improve much more the work, and will undoubtedly facilitate the reader to better understand the effects of oxygen and nitrogen dose and the relationship between them. I believe that you should make this effort to improve the quality of the work and to bring it up to the level of the Plants journal. Of course, the results and discussion should be made based on this analysis. I leave it up to the editor whether or not you should do this analysis for accepting the manuscript for being published.
3. More comments in the pdf text file

Comments on the Quality of English LanguageIn my opinion should be improved
Author Response
|
|||||||||||||||||||||||
